# Chilean Salmon Sushi: Genetics Reveals Product Mislabeling and a Lack of Reliable Information at the Point of Sale

**DOI:** 10.3390/foods9111699

**Published:** 2020-11-19

**Authors:** Valentina Prida, Maritza Sepúlveda, Claudio Quezada-Romegialli, Chris Harrod, Daniel Gomez-Uchida, Beatriz Cid, Cristian B. Canales-Aguirre

**Affiliations:** 1Centro i~mar, Universidad de Los Lagos, Puerto Montt 5480000, Chile; vpridag@gmail.com; 2Núcleo Milenio de Salmónidos Invasores (INVASAL), Concepción 4030000, Chile; maritza.sepulveda@uv.cl (M.S.); claudio.quezada@upla.cl (C.Q.-R.); chris@harrodlab.net (C.H.); dgomezu@udec.cl (D.G.-U.); beatrizcid@udec.cl (B.C.); 3Centro de Investigación y Gestión de Recursos Naturales (CIGREN), Instituto de Biología, Facultad de Ciencias, Universidad de Valparaíso, Playa Ancha, Valparaíso 2340000, Chile; 4Departamento de Biología, Facultad de Ciencias Naturales y Exactas, Universidad de Valparaíso, Playa Ancha, Valparaíso 2340000, Chile; 5Instituto de Ciencias Naturales Alexander von Humboldt, Universidad de Antofagasta, Antofagasta 1271155, Chile; 6Departamento de Zoología, Facultad de Ciencias Naturales y Oceanográficas, Universidad de Concepción, Concepción 4070032, Chile; 7Departamento de Sociología, Facultad de Ciencias Sociales, Universidad de Concepción, Concepción 4070032, Chile

**Keywords:** cyt b, DNA authentication, fish, aquaculture, salmonids, mislabeled, misnamed

## Abstract

Species diagnosis is essential to assess the level of mislabeling or misnamed seafood products such as sushi. In Chile, sushi typically includes salmon as the main ingredient, but species used are rarely declared on the menu. In order to identify which species are included in the Chilean sushi market, we analyzed 84 individual sushi rolls sold as “salmon” from sushi outlets in ten cities across Chile. Using a polymerase chain reaction-restriction fragment length polymorphism protocol (PCR-RFLP), we identified mislabeled and misnamed products. Atlantic salmon was the most common salmonid fish used in sushi, followed by coho salmon, rainbow trout, and Chinook salmon. We found a total of 23% and 18% of the products were mislabeled and misnamed, respectively. In 64% of cases, the salesperson selling the product could not identify the species. We also identified the use of wild-captured Chinook salmon samples from a naturalized population. Our results provide a first indication regarding species composition in Chilean sushi, a quantification of mislabeling and the level of misinformation declared by sales people to consumers. Finally, considering that Chinook salmon likely originates from a non-licensed origin and that sushi is an uncooked product, proper identification in the food production chain may have important consequences for the health of consumers.

## 1. Introduction

The globalization of cuisine has led to rapid and marked changes in how humans consume food, with the consumption of seafood tripling worldwide in the last decade [1], supported by a rapid growth in aquaculture. Human consumption of cultured salmonids (salmon and trout) became increasingly popular [1], especially with regard to salmon, which is seen by many people worldwide as a healthy food choice [2]. Additionally, aquaculture has become a subject of public concern due to issues of environmental impact, human health, and fish welfare [3]. This fuels a demand for alternative seafood networks that offer wild, organic, or fair-trade products [4,5,6]. Although the seafood list published by the U.S. Food and Drug Administration [7] includes guidance regarding acceptable market names to identify particular species, the market names of common and widely consumed species can be vague. This is particularly common among products derived from salmonid fishes, which are often labeled as “salmon”; however, a number of different species of the genera *Oncorhynchus* and *Salmo* fall under this label, showing a lack of species-specific market names for salmonid fishes.

Chile is the second largest aquaculture producer of salmonid fishes worldwide, exceeding 880,000 tons production in 2018 [8,9]. Currently, Atlantic salmon (*Salmo salar*) dominates production in Chile (2018 production: 661,138 tons live weight) followed by coho salmon (*Oncorhynchus kisutch*: 148,521 tons), and rainbow trout (*Oncorhynchus mykiss*: 78,446 tons) [8]. Although the bulk of Chile’s salmonid production is exported, these three cultured species can also be sold nationally following statutory requirements [9]. In addition to aquaculture fish, Chile also supports widespread and abundant populations of naturalized salmonid fishes, where the entire life cycle is completed as free-living individuals [10] including rainbow trout, Chinook salmon (*Oncorhynchus tshawytscha*), and brown trout (*Salmo trutta*) [11,12,13,14]. There is evidence that such naturalized populations could make significant contributions to commercial services supporting human consumption (e.g., Chinook salmon), but the scale or geographical distribution of services for these species provide is unclear. Moreover, at a local level, there is no information available regarding how these species enter the human food chain. As products from non-formal trade are by definition outside of the public health system, they do not have the relevant certifications from health authorities. Although fish originating from naturalized populations may be preferred by some people due to perceived increased quality (e.g., without antibiotics and other chemical treatments), their supply and sale for human consumption is illegal in Chile (except for only Chinook salmon from Toltén River) if they lack any certification from public health authorities regarding their health status or origin.

Correct species identification in the food supply system is crucial as it promotes trust and customer confidence [15]. Chilean food labeling regulations require that species names should be included in ingredient lists (Chilean Ministry of Health). However, this is not always followed by food suppliers. Within the global food sector, seafood from both capture fisheries and aquaculture is associated with the highest rates of mislabeling (i.e., where the product is sold as being from another species or source either intentionally or unintentionally) [16]. A high rate of seafood mislabeling has been identified in restaurant and catering services [17,18]. Intentional mislabeling typically occurs as a fraudulent attempt to pass a product of known lower value as one with higher value [17,19,20]. Mislabeling can also occur unintentionally due to factors not directly related to profit including inadequate or confusing labeling regulations, fisheries targeting similar species, and informal or illicit production chains [21,22]. For instance, the taste and texture of fish flesh can be very similar, making them difficult to discriminate at a species level [23], increasing the likelihood of mislabeling. Additionally, processing removes the morphological characteristics typically used to discriminate between species (i.e., head, skin and fins), and it is almost impossible to identify species from fillets. Such fraud or mis-supply can permanently damage product reputation, especially if human health is impacted by the consumption of fraudulent products mis-sold as genuine [19]. Otherwise, from the point of view of small producers and local communities, although labeling may represent an opportunity to access specialty markets (e.g., organic and fair trade), it also involves important cost and bureaucratic barriers that are difficult to overcome [24]. This results in informal markets where fraud is possible, but can also lead to high-quality local products not being correctly labeled and therefore do not receive the economic premium permitted by the certification processes.

Globally, Asian cuisine is popular with consumers: sushi being one of the most common ways by which seafood is consumed. Originating in Japan, this dish has undergone a massive global boom during the last decade [25], and is particularly popular in Latin America, where recipes have been modified to reflect local materials and consumer preferences. This has resulted in an increase in the number of Japanese restaurants in the region; meanwhile, other non-Japanese restaurants have diversified their menus to include Japanese food [26]. Sushi can include a number of ingredients, but a popular option includes raw fish. The fish species varies by recipe, nonetheless, this type of cuisine is one of the most affected with mislabeling, with salmon, tuna, halibut, and red snapper being the most mislabeled and substituted species [27,28,29]. This has led to a number of different tools being developed to correctly identify the species sold as sushi.

Several techniques exist to allow the identification of species or origin (naturalized versus cultured) including analyses of fatty acids, stable isotopes, carotenoids, and specific protein profiles [30,31,32,33]. However, molecular genetic techniques have many advantages for the identification of processed and unprocessed food as they take advantage of the composition and the conserved regions of the DNA between species and even between populations [34,35,36,37]. A wide variety of DNA-based methods exist for identification including direct sequencing of DNA fragments [38], single nucleotide polymorphisms [39], and digestion of an amplicon by endonucleases [40]. The latter is the least costly and most straightforward to use as it does not require expensive equipment or highly skilled molecular laboratory staff.

In order to identify the species origin of “salmon” sold in the Chilean sushi market, we used a polymerase chain reaction-restriction fragment length polymorphism (PCR-RFLP) protocol on a ~460 bp cytochrome b gene fragment (cyt b) of mitochondrial DNA to be compared with positive controls of salmonids present in Chile (i.e., cultured and naturalized). Specifically, we used the *Dde*I restriction enzyme for species identification given its capacity to reliably produce specific restriction patterns in fishes including salmonids [41,42]. Specific band patterns for *S. salar* (350 and 130 bp), *O. mykiss* (360 and 130 bp), *O. kisutch* (300, 130, and 60 bp), and *O. tshawytscha* (300 and 220 bp) were reported by Russell et al. [40] and Hold et al. [43].

We refer to *mislabeling* where sushi rolls offered on the menu at the point of sale were sold under the generic name of “salmon” (without a species name provided in the menu) but were genetically identified as trout. We refer to Atlantic salmon, coho salmon, Chinook salmon, as salmon, and rainbow trout as trout. We also conducted a small-scale survey to assess knowledge levels regarding salmonid species that are sold in Chile and to identify specific names provided at the point of sale, and these data were subsequently used to estimate the substitution rates. We use the term *misnaming* to refer to any information declared by the salesperson at the point that did not match the molecular identification. We found a total of 23% mislabeled and 18% misnamed products: 64% of salespeople were unable to identify the species they were selling to customers. We also identified the use of wild-captured Chinook salmon samples from a naturalized population. Our results provide a first indication regarding species composition in Chilean sushi, a quantification of mislabeling and the level of misinformation declared by sales people to consumers.

## 2. Materials and Methods

### 2.1. Sample Collection and Survey

Between January 2019 and January 2020, we collected 84 salmonid sushi rolls from premises selling sushi (one roll per each sushi place, i.e., restaurants or take away) from ten different cities located across Chile (Figure 1). Specifically, we collected salmonid sushi roll as samples from five or more different premises selling sushi from each city, except for the city of Quellón. These cities are located along a 3000 km latitudinal gradient extending from the arid north through to the center of salmon aquaculture in the south of Chile. A small sample of uncooked muscle tissue was taken from each sushi roll and frozen at −20 °C. At the point of sale, key characteristics were recorded including the trading name of the premises, purchase date, location, and sushi type: these data are not included further in the analyses reported here. We also asked salespeople what species was provided in the sushi. Finally, we collected samples of the main species of salmonids present and consumed in Chile (Atlantic salmon, coho salmon, Chinook salmon, and rainbow trout), both cultured or naturalized for use as positive controls for genetic analyses.

### 2.2. Molecular Procedures

Genomic DNA of each piece of tissue was extracted using the salting-out protocol [44]. We used the L14735 and H15149 primers described by Burgener [42] to amplify a fragment of the mitochondrial cytochrome b gene (cyt b, ~460 bp). PCR amplifications were performed in a final volume of 50 uL containing 10X Buffer PCR, 25 mM MgCl_2_, 2.5 mM dNTP’s, 10 µM of each primer, 1 U/µL Taq DNA polymerase (Invitrogen^TM^, Carlsbad, CA, USA), and 10 ng/μL of genomic DNA. The amplification program considered a cycle at 94 °C for 5 min, followed by 35 cycles at 94 °C for 20 s, 52 °C for 20 s, and 72 °C for 20 s, and a final extension at 72 °C for 7 min. We followed the protocol for PCR-RFLP analysis described by Hold et al. [43], which included 10X buffer Tango and 3 units of *Dde*I restriction enzyme (Invitrogen^TM^, Carlsbad, CA, USA) incubated at 37 °C overnight. Finally, we visualized the RFLP cutting patterns on a 3.0% agarose gel run in an electrophoresis chamber for 90 min at 95 V under a Safe Imager 2.0 Blue-Light Transilluminator System (Invitrogen^®^, Carlsbad, CA, USA).

### 2.3. Data Analysis

To support the specificity of the restriction enzyme *Dde*I, we conducted an in-silico RFLP using sequences of cyt b obtained from GenBank. We downloaded all sequences that included the cytb gene for each species. We then found and extracted the exact amplicon fragment (~463 bp) flanking the primers L14735 and H15149 in one full mitochondrial genome per species (*S. salar*, JQ390056; *O. kisutch*, EF126369; *O. mykiss*, MT410879; *O. tshawytscha*, NC_002980). We used this amplicon fragment as a reference to align the remaining sequences obtained from GenBank: alignments were conducted using the software MUSCLE v3.8 [45]. During the alignment by species process, sequences larger than 463 were trimmed to this size, sequences shorter than 400 bp were discarded, and we kept a maximum of 20 records per species for visualization purposes. These steps resulted in a final sequence dataset of 71 sequences (Appendix A). Finally, we aligned all fragments for all species included in this study for the in-silico RFLP analysis conducted in CodonCode Aligner (CodonCode Corp., Dedham, MA, USA).

We compared the molecular results obtained by PCR-RFLP in each premises selling sushi with the information declared by salespeople. Samples were considered as mislabeled when the molecular species identification as trout did not match the “salmon” name provided at the point of sale. Additionally, they were considered misnamed when the information declared at the point of sale did not match the molecular identification.

## 3. Results

### 3.1. PCR-RFLP as a Molecular Diagnosis Tool

All samples showed a successful amplification of the cyt b fragment (~460 bp) with unique PCR-RFLP profiles for each salmonid species. The restriction enzyme *Dde*I produced an approximate cut pattern of two bands of 320 and 115 bp for Atlantic salmon, three bands of 260, 115, and 70 bp for coho salmon, two bands of 340 and 115 bp for rainbow trout, and finally two bands of 280 and 160 bp for Chinook salmon (Figure 2). The in-silico RFLP analysis using sequences from GenBank showed the same restriction cut pattern found in our results (Appendix A).

### 3.2. Salmon and Trout Identification, Mislabeling, and Misnaming

All sushi rolls were sold as including “salmon” as a main ingredient, but none identified what species of salmon this referred to. Out of the 84 samples examined, our molecular diagnosis showed that 63% were Atlantic salmon (*n* = 53), 23% rainbow trout (*n* = 19), 10% coho salmon (*n* = 8), and 5% Chinook salmon (*n* = 4). In terms of geographical distribution, sushi containing Atlantic salmon was found in each of the ten different cities sampled. Sushi containing rainbow trout was found in eight of the ten cities, but not in Valdivia or Concepción (Figure 1). Coho salmon was recorded from Puerto Montt and Osorno. Chinook salmon was only identified in sushi from the cities of Temuco and Osorno (Figure 1).

Each of the premises that we visited identified their sushi as containing “salmon” in their menu. As such, we did not expect that samples would include rainbow trout. However, in our sample of Chilean sushi suppliers, we showed a 22.6% mislabeling rate, and this extended to seven of the ten cities we collected samples in (Figure 3). Within a given city, mislabeling rates ranged from 9% (Iquique) to 60% (Valparaiso). When controlled for sample size, mislabeling ranged from 1% (Iquique and Quellón) to 6% (Viña del Mar and Puerto Montt).

With regard to misnaming, 64% (*n* = 54) of staff surveyed at the point of sale were unable to state what species of salmon they were supplying, conversely, 36% (*n* = 30) did provide a species name (Figure 3, Table 1). However, in 50% of such cases (18 of 36), our genetic results showed that they incorrectly identified (i.e., misnamed) the species (Figure 3; Table 1). No sushi supplier identified Chinook salmon as an ingredient of their sushi rolls, while four restaurants stated their product was “salmon-trout” (Table 1, Appendix A), which in some countries is referred to as anadromous brown trout, or sometimes lake trout with a silver phenotype (e.g., United Kingdom) [46].

## 4. Discussion

We were able to successfully identify Atlantic salmon, coho salmon, Chinook salmon, and rainbow trout in the “salmon” sampled from sushi. Similar results have been used to distinguish Atlantic salmon from rainbow trout using PCR-RFLP in mitochondrial (i.e., cyt b, COII, 16S) and nuclear (i.e., 5S) fragments from fresh and smoked samples [47,48,49]. Although restriction enzymes have been widely used to identify salmonids in the food industry [40,49], to the best of our knowledge, it has not been previously used to identify mislabeling and misnaming of species sold as sushi. Russell et al. [40] also used cyt b to identify ten species of salmonids from samples that had been smoked, cooked, and pickled. Later, Hold et al. [43] subsequently showed the utility and reproducibility of the technique across several European laboratories including validated samples. Our results agree with these previous studies using PCR-RFLP for salmonid identification and extend its use to species identification for salmonid sushi.

To date, several PCR-based methods for fish authentication have additionally been used to PCR-RFLP including random amplified polymorphic DNA, single stranded conformational polymorphism, amplified fragment length polymorphism, and direct sequencing of DNA barcoding [50,51,52], with advantages and disadvantages associated with the different approaches [35,53,54]. The recent sequencing revolution has allowed researchers to go to even greater depths with regard to seafood traceability [55], with genomic tools allowing not only the species, but also population to be identified [56,57,58]. Although such new technology has much to offer, our results and recent studies published using PCR-RFLP for traceability in other seafood products [59,60,61,62,63] underline the continued use of PCR-RFLP as a cost-effective, fast, and simple means to identify salmonid species and mislabeling at the point of sale.

Atlantic salmon was the most used species in Chilean sushi, reflecting the dominance of the species both globally and in Chile [8]. The bulk of Chilean production of Atlantic salmon is sold for export [9], with the remainder consumed in the domestic market. In our study, at least 60% of sushi retailers included this species, possibly reflecting a mix of consumer preference for salmon over trout and ease of supply. Among the other three salmonids found in Chilean sushi (coho salmon, rainbow trout, and Chinook salmon), Chinook salmon represents the least expensive option to sushi producers (personal communication by sushi master: Sendai sushi, Temuco, 24 January 2020). At the time of sampling, no Chinook salmon of aquaculture origin were available as aquaculture activities ended in Chile ca. 2009 [12]. As such, Chinook salmon sold in sushi can only have originated from wild-caught individuals from naturalized populations. Chile has several naturalized Chinook salmon populations that spawn in several rivers [12,64,65,66]. However, only one fishery (the Toltén River artisanal fishery) is authorized by the Chilean government to legally capture and sell Chinook salmon [67]. This fishery is located close to Temuco, where the 5% of the total sushi samples included Chinook salmon (four of 84). However, given that salespeople were unaware of the origin of Chinook salmon, it is more likely that the product originated via informal channels as Chinook salmon are also illegally captured and commercialized to catering businesses. Unfortunately, the small fragment sequenced here (~460 bp) to identify species does not have the statistical power to differentiate between populations. However, in the future, genetic studies with a finer resolution could be undertaken to identify the basin of origin of Chinook salmon sold as sushi. To date, two studies at the population level have differentiated individuals from the Toltén River from other rivers in Chile (e.g., single nucleotide polymorphisms, [12]; microsatellites [68]). Use of these two approaches will resolve the issue of which river system the fish originated from, but not whether they were captured in a legal or illegal fishery. It will be important for fishery managers to build an identity for their product that allows it to be recognized in order to counter such problems.

Seafood mislabeling can reflect both fraud or human error in terms of the supply of the product [57] as well as a lack of knowledge at the point of sale. In our sample of Chilean sushi suppliers, the mislabeling rate found was higher than rates published for salmonid products, which ranged from 1% [43] to 11% [69], being more similar to those for non-salmon seafood sales worldwide (e.g., 25%) [29]. In the literature, sushi suppliers have been shown to display higher mislabeling rates (10% to 74%) compared to restaurant and grocery stores [18,27,28,70]. According to Bénard-Capelle et al. [17], the majority of fraud is seen in products purchased from fishmongers, restaurants, and other food outlets. In a Chilean context, mislabeling is due to the substitution of rainbow trout for salmon, at least by Chilean sushi producers. Such issues with mislabeling is also apparent in Chile for non-salmon seafood, such as crustaceans [71] and bivalves [72], even though Chilean labeling regulations require that species names are included in ingredient lists (Chilean Ministry of Health).

Our results highlight a general lack of knowledge by salespersons regarding the species contained in their products. While most of the staff selling sushi were unable to identify a principal ingredient in their product, when an attempt to identify the product was made, it was wrong in half of the cases. There is a clear lack of familiarity with species these outlets are supplying, even from cities where aquaculture is a key economic activity (e.g., Puerto Montt, Quellón; Figure 1).

It is possible that the high mislabeling and misnaming identified here simply reflect a lack of training, interest, or both, by outlet owners and staff at the point of sale. This may result in poor customer service, and may be due to the low wages associated with such work as suggested by Ramírez et al. [73]. We encountered four cases where the product was identified as “salmon-trout”. This term was declared in three locations (i.e., Valdivia, Osorno, and Quellón) and samples were identified as Atlantic salmon and rainbow trout. This type of misnaming results in consumer confusion and supports the idea that staff from sushi outlets are likely unaware of what they are selling and poorly trained on salmonid taxonomy. Confusion in staff from sushi outlets may reflect variation in the products supplied (some weeks, salmon; some weeks, trout), and without training of staff by outlet owners at the point of sale, such confusion can be transferred to consumers. It is likely that this confusion partly reflects the use of the “salmon” label for various species within two distinct genera: *Oncorhynchus* and *Salmo* [69]. However, most people are unaware of this issue, especially so in Chile where salmonids are non-native species.

## 5. Conclusions

This study used PCR-RFLP to identify “salmon” sold in Chilean sushi outlets. Our results showed that menus and statements from salespeople at the point of sale were inaccurate, with high rates of mislabeled (23%) and misnamed (18%) products. We also showed that most salespeople (64%) were unable to identify the species they were selling to customers. Finally, we identified the use of naturalized Chinook salmon in sushi. Given the use of uncooked flesh in sushi, this could represent a potential human health issue if from an illegal source; otherwise, if authorized, formalized, certified, and well identified at the point of sale, this could open a valuable market niche for local, naturalized-caught fish, improving the economic opportunity for fishing communities.

## Figures and Tables

**Figure 1 foods-09-01699-f001:**
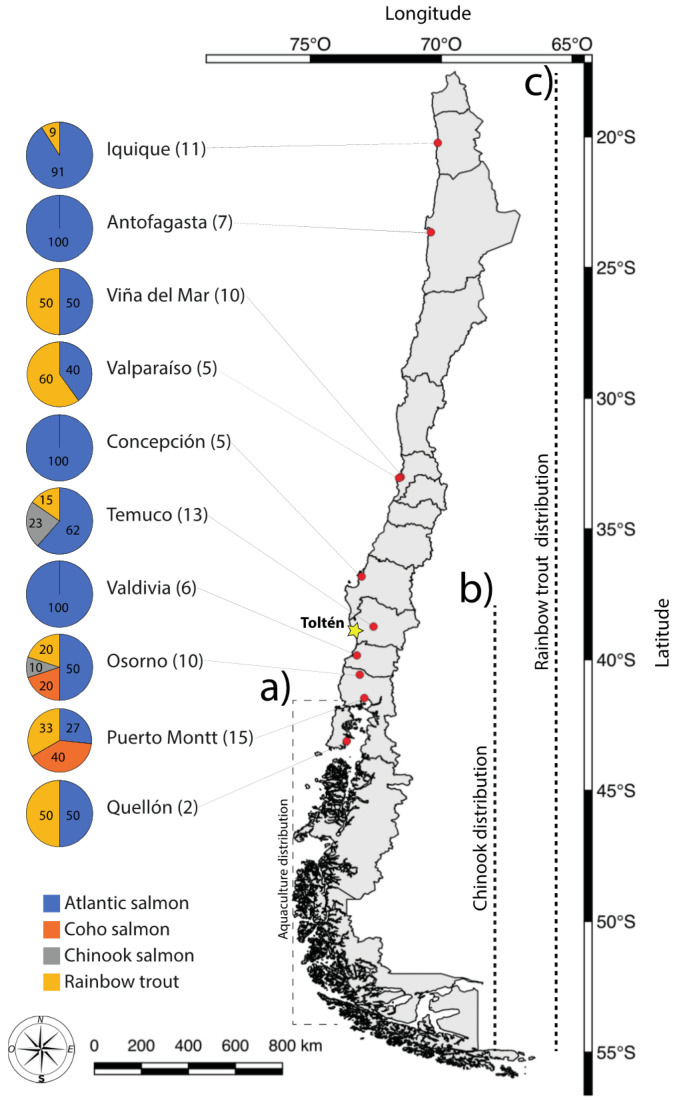
Sampling locations and proportion of sushi samples identified to four different salmonid species based on PCR-RFLP (polymerase chain reaction-restriction fragment length polymorphism). Red markers show the cities where samples were collected, and the yellow marker shows the location of the legal Chinook salmon fishery. Numbers in parentheses reflect the sample size of premises selling sushi in each city, while numbers inside the pie show the percentage. Broken lines show the parts of Chile where (**a**) salmonid aquaculture is common, and (**b**) Chinook salmon and (**c**) rainbow trout have naturalized populations.

**Figure 2 foods-09-01699-f002:**
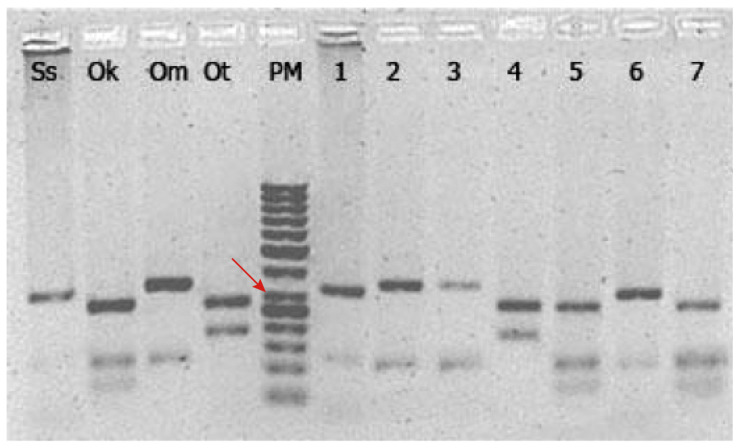
PCR-RFLP analysis of the mitochondrial cytochrome b gene amplified and digested using *Dde*I. Ss: *Salmo salar* (Atlantic salmon), Ok: *Oncorhynchus kisutch* (Coho salmon), Om: *Oncorhynchus mykiss* (Rainbow trout), Ot: *Oncorhynchus tshawytscha* (Chinook salmon), PM: molecular ladder 100 bp. Numbers from 1 to 7 correspond to an arbitrary subset of samples obtained in the sampling collection at different premises selling sushi. The red arrow indicates the 300 bp band.

**Figure 3 foods-09-01699-f003:**
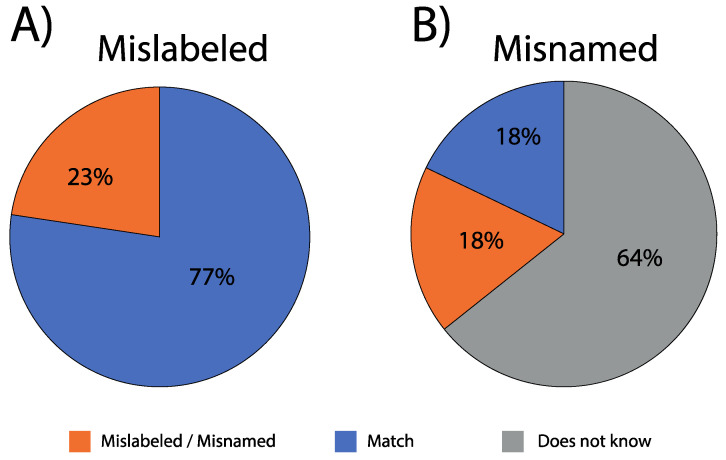
Percentage of Chilean salmonid sushi samples either (**A**) mislabeled (19 of 84) or (**B**) misnamed (15 of 84) when considering all samples collected (*n* = 84).

**Table 1 foods-09-01699-t001:** Number of responses from salespeople to the question “What species was provided in the sushi?” asked by researchers when purchasing ‘salmon’-based sushi in cities across Chile. Species not identified reflects responses where salespeople would/could not provide an answer. Species identification provided reflects the number of cases where salespeople provided information on the species of salmon.

	(A) Salesperson Response		(B) Species Identification Provided					
Locality	Species Not Identified	Species Identification Provided	Atlantic Salmon	Rainbow Trout	Coho Salmon	Salmon-Trout	Species Correctly Named	Species Misnamed
Iquique	11	0					-	-
Antofagasta	6	1	1				1	0
Concepción	5	0					-	-
Viña del Mar	8	2	2				1	1
Valparaíso	3	2	2				1	1
Temuco	11	2	2				2	0
Valdivia	2	4	2		1	1	2	2
Osorno	2	8	6	1		1	5	3
Puerto Montt	6	9	5	1	3		3	6
Quellón	0	2				2	0	2
Total	54	30	20	2	4	4	15	15

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
