# Peer review of "Chilean Salmon Sushi: Genetics Reveals Product Mislabeling and a Lack of Reliable Information at the Point of Sale"

_foods, 2020, doi:10.3390/foods9111699_

Round 1

Reviewer 1 Report

L.26: commonlycommon

L.46: duplicate periods

L.147: use lower case for cytochrome

L.148: MgCl2

L.152: I of DdeI must not be in italic. Same for L.166, 172

L.174: bp

There is no background data to show how RFLP analysis by DdeI restriction enzyme is reliable. Is there no intraspecific RFLP in each species? There must be many cyt b data of salmonid species in database, so the author may be able to confirm it. Or, if the author have evaluated the reliability of the DdeI marker previously, the number of individuals analyzed could be presented.

Author Response

We would like to thank the the reviewer for her/his rapid and useful comments. We have responded to her/his and included updated text below.

Reviewer 1

  1. There is no background data to show how RFLP analysis by DdeI restriction enzyme is reliable.

Author response: Thank you for pointing this out. We have added relevant background information to show that RFLP analysis by DdeI restriction enzyme is reliable in this case.

The revised text reads as follows on page 3 lines: 119-123:

“Specifically, we used the DdeI restriction enzyme for species identification given its capacity to reliably produce specific restriction patterns in fishes including salmonids [39,40]. Specific band patterns for S. salar (350 and 130 bp), O. mykiss (360 and 130 bp), O. kisutch (300, 130, and 60 bp), and O. tshawytscha (300 and 220 bp) were reported by Russell et al. [38] and Hold et al. [41].”

  1. Is there no intraspecific RFLP in each species? There must be many cyt b data of salmonid species in database, so the author may be able to confirm it. Or, if the author have evaluated the reliability of the DdeI marker previously, the number of individuals analyzed could be presented.

Author response: We thank the reviewer for this excellent suggestion to confirm the reliability of this enzyme using data from public databases. As such, we have downloaded sequences for S. salar, O. kisutch, O. mykiss, and O. tshawytscha that fit the length criterium (~450 bp). We applied an in silico RFLP using DdeI to test specificity and we built a virtual gel. We included this information in Materials and methods section on page 5 lines: 174-184, Results section on page 5 lines: 197-198, and a Supplementary Table (Table S1) and Figure (Figure S1).

The revised text reads as follows on page 5 lines: 174-184:

“To support the specificity of the restriction enzyme DdeI we conducted an in-silico RFLP using sequences of cyt b obtained from GenBank. We downloaded all sequences that included the cytb gene for each species. We then found and extracted the exact amplicon fragment (~463 bp) flanking the primers L14735 and H15149 in one full mitochondrial genome per species (S. salar, JQ390056; O. kisutch, EF126369; O. mykiss, MT410879; O. tshawytscha, NC_002980). We used this amplicon fragment as a reference to align the remaining sequences obtained from GenBank: alignments were conducted using MUSCLE v3.8 [43]. During the alignment by species process, sequences larger than 463 were trimmed to this size, sequences shorter than 400 bp were discarded, and we kept a maximum of 20 records per species for visualization purposes. These steps resulted in a final sequence dataset of 71 sequences (Table S1). Finally, we aligned all fragments for all species included in this study for the in-silico RFLP analysis conducted in Codoncode Aligner (CodonCode Corporation).”

… on page 5 lines: 197-198:

“The in-silico RFLP analysis using sequences from GenBank showed same restriction cut pattern found in our results (Figure S1)”

Reviewer 2 Report

The article entitle “Chilean Salmon Sushi: Genetics reveals Product mislabeling and a lack of reliable Information at the Point of Sale” is about an interesting topic- food frauds- notwithstanding I think that the contribution is not so relevant for the scientific community. Furthermore from the point of view of study design it is not clear how restaurant have been choosen and if the sample size used in text is representative of the entire Chile The method used (RFLP) is known and only its application to distinguish salmon is novel. I think that RFLP is not so specific and I would like to know if authors have confirmed the validity of the methods before using it in the survey. Finally I would like to know if the three Chinook salmon identified have been confirmed with other techniques (sequencing). The results report % of mislabeled and misnamed products, but it is not clear the validity of results anf it is not reported the Interval confidence of value No minor comment are necessary Marzia Pezzolato DVM PhD

Author Response

We would like to thank the Dra. Pezzolato for her rapid and useful comments. We have responded to her and included updated text below.

Reviewer 2

  1. The article entitle “Chilean Salmon Sushi: Genetics reveals Product mislabeling and a lack of reliable Information at the Point of Sale” is about an interesting topic- food frauds- notwithstanding I think that the contribution is not so relevant for the scientific community.

Response 1: While we appreciate the Dra. Pezzolato’s feedback, we respectfully disagree.

We think that perceived novelty or relevance is not the only purpose for doing science and this study makes a valuable contribution to this field considering that salmonids are used in several dishes including sushi as we show in this study. Sushi is one of the most commonly consumed fast foods in Chile. Our results provide a first indication regarding species composition in Chilean sushi sold as salmon, reveal (and quantify) mislabeling and the level of misinformation declared by salespeople to consumers. Additionally, we identified Chinook salmon in the food chain, which, given its likely origin from a non-licensed source, and that sushi is uncooked, has potential consequences for consumer health.

  1. Furthermore from the point of view of study design it is not clear how restaurant have been choosen and if the sample size used in text is representative of the entire Chile

Response 2: The study was designed to i) extend across locations where the population are either less (e.g. Iquique and Antofagasta) or more familiar with salmonid fishes (e.g. Puerto Montt, Osorno), and ii) trying to maximize the number of sushi point of sale locations in each city. Although we were not able to sample form all locations selling sushi, we feel that the sample size is representative for mostly of the cities.

  1. The method used (RFLP) is known and only its application to distinguish salmon is novel.

Response 3: We are agreed with Dra. Pezzolato’s comment and appreciate her observation regarding the novelty of extending the RFLP approach to identifying salmonids used to prepare sushi.

  1. I think that RFLP is not so specific and I would like to know if authors have confirmed the validity of the methods before using it in the survey.

Response 4: Dra. Pezzolato’s assessment was echoed by Reviewer 1. Following these observations, we included i) background information showing how RFLP analysis by DdeI restriction enzyme is reliable for the identification of the species used in this study, and ii) an additional analysis to confirm the reliability of this enzyme using sequences from GenBank displaying the cut pattern in an in-silico RFLP gel. The revised text reads as follows on page 3 lines: 119-123; on page 5 lines: 174-184; and on page on page 5 lines: 197-198:

The revised text reads as follows on page 3 lines: 119-123:

“Specifically, we used the DdeI restriction enzyme for species identification given its capacity to reliably produce specific restriction patterns in fishes including salmonids [39,40]. Specific band patterns for S. salar (350 and 130 bp), O. mykiss (360 and 130 bp), O. kisutch (300, 130, and 60 bp), and O. tshawytscha (300 and 220 bp) were reported by Russell et al. [38] and Hold et al. [41].”

The revised text reads as follows on page 5 lines: 174-184:

“To support the specificity of the restriction enzyme DdeI we conducted an in-silico RFLP using sequences of cyt b obtained from GenBank. We downloaded all sequences that included the cytb gene for each species. We then found and extracted the exact amplicon fragment (~463 bp) flanking the primers L14735 and H15149 in one full mitochondrial genome per species (S. salar, JQ390056; O. kisutch, EF126369; O. mykiss, MT410879; O. tshawytscha, NC_002980). We used this amplicon fragment as a reference to align the remaining sequences obtained from GenBank: alignments were conducted using MUSCLE v3.8 [43]. During the alignment by species process, sequences larger than 463 were trimmed to this size, sequences shorter than 400 bp were discarded, and we kept a maximum of 20 records for species for visualization purposes. These steps resulted in a final sequence dataset of 71 sequences (Table S1). Finally, we aligned all fragments for all species included in this study for the in-silico RFLP analysis conducted in Codoncode Aligner (CodonCode Corporation).”

… on page 5 lines: 197-198:

“The in-silico RFLP analysis using sequences from GenBank showed same restriction cut pattern found in our results (Figure S1)”

  1. Finally I would like to know if the three Chinook salmon identified have been confirmed with other techniques (sequencing).

Response 5: We have not confirmed this using other technique such as through sequencing, mainly because we want to reproduce a cost-effective, fast, and simple protocol. However, as noted in response 4, the restriction pattern obtained for the three Chinook individuals is similar to that obtained from the in-silico RFLP.

  1. The results report % of mislabeled and misnamed products, but it is not clear the validity of results anf it is not reported the Interval confidence of value

Response 6: We feel that after confirming the reliability of this enzyme using sequences from GenBank, our results are validated. We cannot include any confidence intervals as our results are showed as percentages.

  1. No minor comment are necessary Marzia Pezzolato DVM PhD

Response 7: Thank you for your useful comments.

Reviewer 3 Report

The article deal with the survey made in Chile sampling in sushi resturants or corner of which salmonid fish is present in sushi and if there is a correspondance between the identified species and what is declared by labels or salesman. 

The topic could be of interest for the Chillean community or fot the local authorities that must controll the correspondence of the information given to consumes and what is present in the resturants dishes. There is some little more efforts than authors must do. 

First there is a need to increase the quality of english. 

One other consideration regards the possibility to identify flesh of farmed atlantic salmon, with a clear presence of fat line between myomers, and flesh of rainbow trouth or wild pacific salmon. Furthermore there there where not differences in color between species? A farmed Atlantic salmon usually could easly recognized and distinguished from wild pacific salmons by a color analysis. Author did not notice any differences between them? 

In all the manuscript there is a distinction of mislabelling and misnaming but it is not clear what difference there is between the two misidentification? Why authors underline the two mistakes? One is more serious than the other by a chillean legal point of view? 

Salmon-trout is the common name used to identify a trout where carotenoid are added to the diet to cange the colour of trout fillet to resemble salmon in color. 

Other small comment are reported in the attached file. 

Author Response

We would like to thank the the reviewer for her/his rapid and useful comments. We have responded to her/his and included updated text below.

Reviewer 3

  1. The article deal with the survey made in Chile sampling in sushi resturants or corner of which salmonid fish is present in sushi and if there is a correspondance between the identified species and what is declared by labels or salesman. The topic could be of interest for the Chillean community or fot the local authorities that must controll the correspondence of the information given to consumes and what is present in the resturants dishes. There is some little more efforts than authors must do.

Response 1: We hope that our responses to the comments suggested by the yourself and the other two reviewers have helped improve the manuscript.

  1. First there is a need to increase the quality of english.

Response 2:  One of the authors (Chris Harrod) is a native speaker from the UK. We are confident that the text is readable and understandable.

  1. One other consideration regards the possibility to identify flesh of farmed atlantic salmon, with a clear presence of fat line between myomers, and flesh of rainbow trouth or wild pacific salmon. Furthermore there there where not differences in color between species? A farmed Atlantic salmon usually could easly recognized and distinguished from wild pacific salmons by a color analysis. Author did not notice any differences between them?

Response 3:  Thank you for this interesting suggestion that we will examine in future work, and that has clear potential as an alternative means to assess origin. In our study, we did not examine differences in flesh color or identify fat lines between myomeres:  as such, we cannot include this information in the revised version of this manuscript. One further issue is that farmed salmon come in a variety of colors (which are defined by the choice of feed that reflects different market demands), so we don't think a visual inspection of parts that are already processed (i.e. hot sushi rolls) would be sufficient to discriminate between species.

  1. In all the manuscript there is a distinction of mislabelling and misnaming but it is not clear what difference there is between the two misidentification? Why authors underline the two mistakes? One is more serious than the other by a chillean legal point of view?

Response 4:  We understand that can be hard to differentiate between mislabeling and misnaming, which although appear similar, are slightly different.

Put simply, mislabeling reflects the label provided on the menu at the point of sale, while misnaming reflects the name given by the salesperson at the point of sale. We included this information in Introduction section on page 3 lines: 124-126, and 129-131

The revised text reads as follows on page 3 lines: 124-126:

“we refer to mislabeling where sushi rolls offered on the menu at the point of sale that were sold under the generic name “salmon” (without a species name provided in the menu) but were genetically identified as trout.”

The revised text reads as follows on page 3 lines: 129-131:

 “We use the term misnaming to refer to any information declared by the salesperson at the point of sale that did not match with molecular identification.”

  1. Salmon-trout is the common name used to identify a trout where carotenoid are added to the diet to cange the colour of trout fillet to resemble salmon in color.

Response 5:  This is an extremely interesting aspect to explore for future studies. We are unaware of any published reference to this in S America or Europe and would be very appreciative if the reviewer could share some relevant references on this topic.

  1. Other small comment are reported in the attached file.

Response 6: Unfortunately, there was no attached file. We appreciate all comments pointed out for this manuscript.

Round 2

Reviewer 2 Report

The paperchas been improved as request .

Major detail on methods have been inserted

Author Response

Thanks for providing valuable comments on earlier drafts of the manuscript.